



# Soil disturbance in wetlands by feral pigs increases greenhouse gas emissions

Maria Fernanda Adame[1], Alex Pearse[2], Jack W Hill[2], Swade Cristo[3], Jonathan Nadji[4], Jasmine Malua Hall[1], Jessa Thurman[1,5], Valerie Hagger[2], Catherine E Lovelock[2]

[1]Australian Rivers Institute, Griffith University, QLD, 4111, Australia

[2] School of Environment, The University of Queensland, St Lucia, 4072, Australia

[3]Murrumburr clan, Kakadu National Park, NT, Australia

[4]Bunitj clan, Kakadu National Park, NT, Australia

[5]Agriculture & Food, CSIRO, Dutton Park, QLD, 4102, Australia

*Correspondence to*: MF Adame (f.adame@griffith.edu.au)

**Abstract.** Multiple approaches are needed to decrease greenhouse gas emissions and reduce the pace of climate change. Wetlands are among the most carbon-rich ecosystems on the planet, and when disturbed, they can generate disproportionately high emissions. Invasive hoofed mammals, such as feral pigs (*Sus scrofa*), cause significant soil disturbances by trampling,

grubbing, and digging in wetlands. We tested whether soil disturbances by feral pigs would increase greenhouse gas emissions (carbon dioxide, $CO_2$, methane, $CH_4$, and nitrous oxide, $N_2O$) as a result of reduced soil oxygen and plant cover, and increased nitrogen. Six paired sites were sampled in Kakadu National Park, northern Australia, a site of immense cultural and natural significance. Fluxes of $CH_4$ were significantly different among treatments, with emissions being higher at disturbed plots (663 ± 740, 54 to 4,820 µg m$^{-2}$ hr$^{-1}$) compared to reference plots (375 ± 292, -8.6 to 1,785 µg m$^{-2}$ hr$^{-1}$). The most notable differences

were observed for $N_2O$, with significantly higher emissions at disturbed plots (81 ± 88.7, 26.6 to 548 µg m$^{-2}$ hr$^{-1}$), which were up to an order of magnitude higher than those for the reference plots (11.9 ± 3.0, 2.7 to 20.4 µg m$^{-2}$ hr$^{-1}$). Soil redox values were correlated with emissions in plots disturbed by pigs, with negative values associated with high $CH_4$ emissions. The highest emissions were found in recently disturbed sites. This study provides another compelling example of how animal populations can significantly impact the carbon cycle at the landscape scale. It also provides evidence for the viability of a carbon

methodology to reduce greenhouse gas emissions through feral pig management, which will support both culture and nature.

## 1 Introduction

As climate change and human impacts on landscapes intensify, multiple approaches are needed to decrease greenhouse gas emissions. First and foremost, reducing fossil fuel use is required; additionally, nature-based solutions that reduce emissions while increasing ecosystem and societal benefits are possible (Matthews et al., 2022). One such solution is the restoration and

improved management of wetlands.

Wetlands are one of the most carbon-rich ecosystems on the planet; their high soil carbon content and low oxygen conditions favour sequestration but can also lead to potentially high methane ($CH_4$) and nitrous oxide ($N_2O$) emissions. These potent



greenhouse gases are emitted in large quantities in wetlands from tropical climates, mostly when they are polluted (Rosentreter
et al., 2021; Saunois et al., 2020). Emissions from wetlands increase when soils are disturbed, for example, by the trampling,
grubbing, and digging of invasive ungulates or hoofed mammals (O'Bryan et al., 2021). Thus, managing invasive species
could decrease greenhouse gas emissions and improve wetland carbon sequestration (Rowland & Lovelock, 2024; Schmitz et
al., 2014).

In tropical Australia, the management of invasive ungulates, such as water buffaloes (*Bubalus bubalis*), horses (*Equus*
*caballus*), and feral pigs (*Sus scrofa*), is a huge challenge (Bradshaw et al., 2007a). Pigs are especially abundant, with the
latest estimates of around 3.2 million pigs throughout Australia, with a mean density of one pig per square kilometre (Hone,
2020). Feral pigs are common in wetlands, where they cause extensive damage to soils, disperse weeds, and prey on native
species (Ballari and Barrios-García, 2014; Waltham and Schaffer, 2018). Introducing feral pigs into Australia has had
catastrophic effects on biodiversity and ecosystem functions. For instance, pigs consume large numbers of native animals,
such as freshwater turtles (Doupé et al., 2009). Meanwhile, feral pigs have become prey to native saltwater crocodiles
(*Crocodylus porosus*), which currently obtain a significant portion of their nutrition from these invasive animals (Adame et
al., 2018). Finally, feral pigs are potential carriers of human diseases such as Japanese encephalitis, which is transmitted by
mosquitoes (Bradshaw et al., 2007a). To date, controlling the numbers of feral ungulates has been intermittent and
controversial, as some people consider them an important food source (Bradshaw et al., 2007). Long-term planning with cost-
effective strategies is necessary to effectively mitigate the environmental and human impacts of feral pigs (Hamnett et al.
2024).

Carbon markets can provide long-term funding for projects that reduce greenhouse gas emissions, increase carbon
sequestration, and protect carbon stocks. In Australia, a national carbon market exists, allowing for the generation of credits
(Australian Carbon Credit Units, ACCUs) from projects in agriculture, energy, landfill and waste management, mining,
transport, and vegetation (Clean Energy Regulator, CER, 2024). The latter includes nature-based projects such as vegetating
landscapes, restoring tidal wetlands, and managing fires (CER, 2024). A methodology that reduces greenhouse gas emissions
by managing invasive ungulates could provide cultural and economic benefits while reducing environmental impacts.
However, the emissions from pig disturbance of wetlands are still uncertain (median of 4.9 million $MtCO_2$ per year and a wide
range from -30.3 to 94 million $MtCO_2$), and further investigations assessing the climate benefits of feral pig management are
required (O'Bryan et al., 2022).

This study was conducted in Kakadu National Park in northern Australia (Fig. 1). The park has immense cultural and natural
values, being one of the last unmodified tropical rivers in the world. Kakadu has been home to First Nations People (Traditional
Owners) for more than 65,000 years (Clarkson et al. 2017), being the oldest living culture on Earth with a strong spiritual
connection to "country", which refers to the lands, waterways, and seas (physical, cultural, and spiritual) with which they are



associated. The Traditional Owners of Kakadu are the Bininj in the north and Mungguy in the south, and together, they manage
and lead research that is connected within the park (Robinson et al., 2022).

We measured the effect of soil disturbance in wetlands by feral pigs on greenhouse gas emissions (carbon dioxide, $CO_2$, $CH_4$,
and $N_2O$) by comparing 1) six paired plots distributed across two floodplains with and without soil pig disturbance, 2)
variability from time of disturbance (recently and old disturbed), and 3) variability within two seasons: Yekke, the cool post-
monsoon season, and Kunumeleng, the dry and hot pre-monsoon season. We hypothesised that pig disturbance in wetlands
would result in changes in the physicochemical properties of the soil. We predict that soil nitrogen and labile carbon will
increase due to animal excretion (Krull et al., 2013), and that soil oxygen (and thus its reduction-potential or redox) will
decrease due to the increased microbial respiration and the death of fine roots (Doupé et al., 2009; Liu et al., 2020). We expect
these changes to result in increased incomplete nitrification-denitrification, which increases $N_2O$ emissions (Anderson &
Levine, 1986) and an increase in soil decomposition, favouring $CO_2$ and $CH_4$ emissions. Finally, we predict that emissions
would be highest immediately after disturbance (Liu et al., 2020), especially during the hotter Kunumeleng season, as higher
temperatures accelerate $CH_4$ production (Yvon-Durocher et al., 2014).

## 2 Methods

### 2.1 Study site and sampling design

Kakadu National Park is bounded by the Van Dieman Gulf to the north, and the Kakadu escarpment to the south; three main
rivers flow through the park, Wildman River, and South and East Alligator (Fig. 1). The region has a monsoonal climate with
a total mean rainfall of 1,547 mm, mostly falling between December and March (1,383 mm); annual mean temperature ranges
between 22.7 and 34.4°C (1971-2021; Bureau of Meteorology, BM, Jabiru Airport, Station 14198). The climate is classified
into six seasons according to traditional knowledge: Kudjewk or monsoon season (December to March), Bangkerreng or storm
season (April), Yekke or cool and humid season (May to mid-June), Wurrkeng or cold season (mid-June to August), Kurrung
or hot dry season (mid-August to October), and Kunumeleng or pre-monsoon storm season (mid-October to late December).
During the Kudjewk, the rivers that run through Kakadu overflow and inundate adjacent low-lying areas where sediments
carried by the river are deposited, and dense vegetation grows (Adame et al., 2017; Erskine et al., 2018). These floodplains
can stay inundated for six to eight months of the year (Ward et al., 2016). During the dry season, flooding recedes, and
permanent deep-water holes remain in the floodplain until the next wet season, when they are once again reconnected to the
river channel





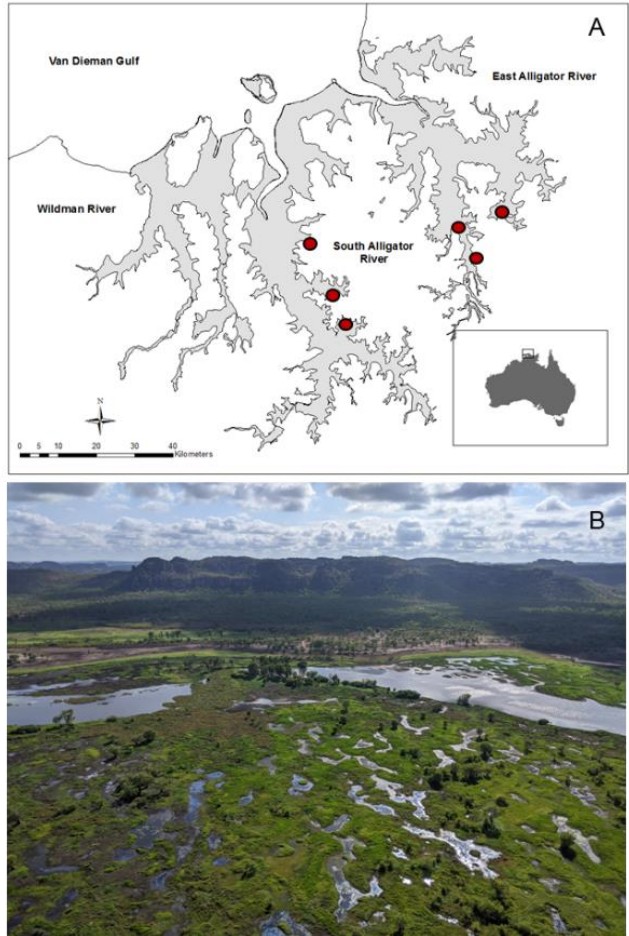

**Figure 1. (A) Location of Kakadu National Park in northern Australia, and sampling points (north to south): Munmarlay, Coolaboo, Mamukala in the South Alligator and Ubirr, Djabulukgu and Dja Dja in the East Alligator, (B) floodplain wetlands of the Magela**

**Creek adjacent to the East Alligator River. Picture by MF Adame.**

We conducted two field campaigns, one during Yekke (8-14th June 2023) or the cool season and one during Kunumeleng (25-30th October 2023) or the hot, pre-monsoon season. Sampling in other seasons was not possible due to the remoteness of the area, extreme heat, and the dangers posed by saltwater crocodiles (*Crocodylus porosus*) during the wetter seasons of the year. For our first trip, six sites were selected by the Traditional Owners co-authors. The selected sites were Munmarlay, Coolaboo

and Mamukala in the South Alligator and Ubirr, Djabulukgu and Dja Dja in the East Alligator (Fig. 1).

The vegetation of the sampling sites included floodplain grassland with scattered broad-leaved paperbark (*Melaleuca viridiflora*) at Dja Dja and Ubirr, paperbark forests at Djabulukgu, Coolaboo, and Munmarlay (*M. viridiflora, Melaleuca cajuputi*), and freshwater mangroves (*Barringtonia acutangula*) at Mamukala. Away from the open water, the wetlands were





surrounded by savanna or pandanus (*Pandanus spiralis*) woodland. We sampled a reference plot at each site, either grassland
or a forest of paperbark forest which was relatively undisturbed and a plot nearby with soil disturbances (Fig. 2). The disturbed
plots were either "digging areas" that pigs use for finding roots and other underground food (e.g., turtles) or "wallows" which
were deeper and are used by pigs to cover themselves with mud to avoid overheating (Bracke, 2011). Although the soil
disturbances were mostly done by pigs, feral water buffalo (*Bubalus bubalis*) may have also contributed, as buffalo wallows
and tracks were observed at Dja Dja, Ubirr, and Munmarlay.

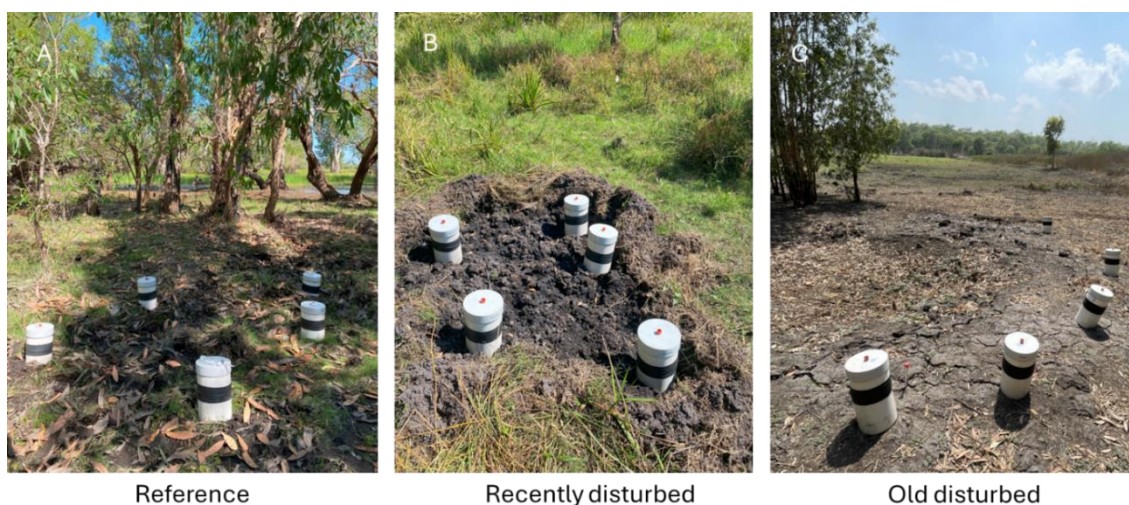


**Figure 2. Examples of (A) a reference or undamaged plot at Djabulukgu, (B) a recently disturbed plot by feral pigs at Coolaboo, and
(C) an old, disturbed plot that was bare of vegetation and dry (Djabulukgu). Pictures by JM Hall.**

On our second trip during the Kunumeleng season, we resampled three of the sites (Munmarlay, Djabulukgu, and Mamukala).
But this time, targeting not only reference and disturbed plots but also areas that were previously disturbed and were now
unvegetated and dry (Fig. 2). The three treatments (reference, recently disturbed and old disturbed) were as close to each other
as possible, usually less than 50 m away, except for our plot in Munmarlay, where we could not access a recently disturbed
plot, and was paired with a reference plot in nearby Coolaboo (Fig. 1).

## 2.2 Soil physicochemical characteristics

At each plot, reduction-oxidation potential (redox; 3-5 measurements) and soil temperature (five measurements) were
determined with a redox meter (HQ 11d ORP- meter, Hach). Redox from reference and old disturbed sites during the
Kunumeleng season could not be measured as the soil was completely dry. Three soil samples of a known volume were taken



at each plot, weighed, and oven-dried at 50˚C. They were then weighed again, from which the water content and bulk density were calculated. The samples were analysed for organic carbon (after testing for inorganic carbon, C, through HCl addition),

nitrogen (N), $\delta^{13}C$, and $\delta^{15}N$ (EA-IRMS, Serco System at Griffith University). The particle size distribution of surface sediments (0-5 cm deep) was assessed from three subsamples from each plot, which were homogenised into a single sample and sieved through meshes of the following sizes: < 20 µm (fine silt and clay), >20 µm (medium and coarse silt) ≤, > 63 µm (very fine sand) ≤, >125 µm (fine, medium and coarse sand) ≤, and > 2 mm (gravel). The contents of each sieve were weighed, and their contribution to the total sample was calculated.  Finally, soil conductivity was measured with a conductivity meter

(ProDSS YSI, OH, USA or HACH) from a solution of soil to water (1:5) adjusted to the dominant grain size class of each plot.

**2.3 Greenhouse gas emissions**

Instantaneous gas fluxes (n = 5 per plot) were measured in dark, closed chambers (Hutchinson & Mosier, 1981) made of polyvinyl chloride (4L, 15 cm diameter and 30 cm in height; Fig. 2). During each experiment, gas samples were collected from the headspace of the chambers with a syringe at 0, 20, 40, and 60 min and transferred into vacuumed containers (Labco, High

Wycombe, UK; Brannon et al., 2016). The $CO_2$, $CH_4$, and $N_2O$ concentrations within each sample were measured in a gas chromatograph (Shimadzu GC-2010 Plus, DESITI, Queensland Government). Linearity of fluxes within each chamber in time was checked. Those who complied were used to calculate fluxes (*F*) per hour from the net changes in gas concentrations (*N*, mol) during the incubation time (*t*, hours), per volume of the chamber (*V*, m3), at a given temperature of the soil (*T*, °K), assuming a pressure of one atmosphere:

$$F = \frac{N}{RT} \times \frac{V}{t\,A}$$

Equation 1

where,
$R = 8.205 \times 10\text{-}5$ m3 atm-1 K-1 mol-1

**2.4 Root versus soil respiration**

Soil $CO_2$ fluxes are a combination of the decomposition of organic matter and dead roots and the respiration of live roots, which can contribute up to 79% of the total (Krauss et al. 2012).  To test the contribution of root respiration to soil $CO_2$ emissions, we measured instantaneous gas fluxes and biomass from roots in reference and disturbed plots by sampling sediment cores at each plot. On the same collection day, the sediments were washed and sieved through a 1 mm mesh, and roots were

hand-picked, blotted dry, and weighed. The $CO_2$ flux from each pooled sample (coarse and fine roots) was measured in the



dark with a portable gas analyser (Picarro, GasScouter, California, USA). The following week, back in the laboratory, the roots were oven-dried at 60°C and reweighed to measure their biomass.

## 2.5 Emission reduction potential

To estimate the carbon abatement potential of feral pig management, $CH_4$ and $N_2O$ fluxes were converted to $CO_{2eq}$ by
multiplying them by their 100-year warming potential, which is 27 for $CH_4$ (non-fossil origin) and 273 for $N_2O$ (IPPC, AR6 Sixth Assessment Report, 2021). Fluxes of $CO_2$ were also included by correcting for respiration using the differences in emissions (%) from vegetated and unvegetated plots. The rates were extrapolated to annual fluxes per hectare.

## 2.6 Statistical analyses

Data was tested for normality with Shapiro-Wilk and Kolmogorov-Smirnov tests, for datasets that complied with normality, a two-way nested ANOVA with main effects (disturbance: old/new/reference) nested within the site was conducted to test differences between disturbed (recently and old disturbed) and reference plots across the six sites for gas fluxes ($CO_2$, $CH_4$ and $N_2O$) and physicochemical parameters (soil conductivity, temperature, redox, soil C, N, C:N, $\delta^{15}N$ and $\delta^{13}C$). For datasets that did not comply with normality ($CH_4$ fluxes), comparisons among treatments were conducted with a non-parametric Wilcoxon
matched-pairs signed-rank test. A repeated-measurements ANOVA was used to test for differences in gas fluxes between the two seasons. Regression and non-linear model fits were conducted to assess the response of greenhouse gas fluxes to physicochemical parameters. Statistics were analysed with GraphPrims 10.2.3. Values are mean ± standard error (min-max) unless specified otherwise.

## 3 Results

### 3.1 Soil physicochemical characteristics

During Yekke (cool season), soil conductivity, temperature, and redox were significantly higher at the reference plots (180 ± 74.3 µS cm$^{-1}$, 27.8 ± 1.7°C, 206.3 ± 53.6 mV) compared to the disturbed plots (56.7 ± 7.6 µS cm$^{-1}$, 25.3 ± 0.8°C, 67.7 ± 15.5 mV; $F_{1,24}$ = 580.6 $p < 0.001$, $F_{1,47}$ = 424.7 $p < 0.001$ and $F_{1,24}$ = 26.19 $p < 0.001$, respectively; Table 1). In contrast, bulk density
and water content were similar between treatments (58.0 ± 4.6 % and 0.46 ± 0.05 g cm$^3$ versus 53.2 ± 5.2 % and 0.45 ± 0.08 05 g cm$^{-3}$ for reference and disturbed plots, respectively; Table 1).



**Table 1. Soil physicochemical characteristics of six reference plots paired with six recently disturbed plots by feral pigs during Yekke (cool season). Soil conductivity, temperature, and redox were significantly higher at the reference compared to the disturbed plots** 180 **($F_{1,24}$ = 513.2 $p$ < 0.01, $F_{1,47}$ = 424.7 $p$ < 0.01, and $F_{1,24}$ = 26.19 $p$ < 0.001). Significant differences are shown as $p$ < 0.05* and $p$ < 0.001**.**

| | Conductivity (µS cm$^{-1}$) | | Redox (mV) | |
| --- | --- | --- | --- | --- |
| | Reference | Disturbed | Reference | Disturbed |
| Ubirr | 239 ± 15 | 103 ± 4** | 125.3 ± 13.2 | 79.9 ± 44.5 |
| Munmarlay | 532 ± 35 | 98 ± 2** | 174.6 ± 17.9 | 22.0 ± 68.6* |
| Mamukala | 1,319 ± 59 | 216 ± 12** | 445.7 ± 5.6 | 89.5 ± 42.9** |
| Coolaboo | 60 ± 2 | 80 ± 3 | 61.3 ± 15.6 | 16.9 ± 58.4 |
| Djaja | 126 ± 7 | 103 ± 10 | 210.8 ± 90.8 | 99.2 ± 67.1 |
| Djabulukgu | 128 ± 3 | 65 ± 3* | 220.0 ± 22.0 | 98.5 ± 21.4 |
| | Temperature (°C) | | Water content (%) | |
| | Reference | Disturbed | Reference | Disturbed |
| Ubirr | 35.0 ± 0.0 | 24.0 ± 0.0** | 43.8 ± 1.8 | 39.2 ± 3.6 |
| Munmarlay | 24.5 ± 0.0 | 26.0 ± 0.0** | 61.7 ± 2.8 | 54.5 ± 2.8 |
| Mamukala | 24.7 ± 0.0 | 24.0 ± 0.0* | 73.4 ± 1.4 | 73.5 ± 1.2 |
| Coolaboo | 31.0 ± 0.5 | 29.0 ± 0.0** | 48.9 ± 1.6 | 61.1 ± 1.9 |
| Dja Dja | 26.0 ± 0.0 | 24.0 ± 0.0** | 67.0 ± 1.3 | 49.3 ± 1.1 |
| Djabulukgu | 25.6 ± 0.4 | 25.0 ± 0.0* | 53.5 ± 1.2 | 41.9 ± 1.6 |

Surface soil C and N, and molar C:N were also similar between reference (8.1 ± 1.1% C and 0.63 ± 0.09% N, 9.7 ± 0.9) and disturbed plots (8.5 ± 1.4% C and 0.68 ± 0.10% N, 10.8 ± 0.7; $p$ = 0.480, $p$ = 0.312, and $p$ = 0.251, respectively). Soil $\delta^{15}$N did not differ between treatments, with 2.73 ± 0.60‰ for reference compared to 2.74 ± 0.17‰ for disturbed plots (Table 2). 185 However, $\delta^{13}$C was slightly but significantly higher at the reference plots (-22.6 ± 1.7‰) compared to the disturbed ones (-21.4 ± 1.7‰; $F_{1,24}$ = 8.66, $p$ = 0.007; Table 2).

All sites had different sediment grain size composition; Mamukala, Dja Dja and Djabulukgu were dominated by sand (fine, medium and coarse), Ubirr was dominated by silt (medium and coarse), while Munmarlay and Coolaboo were dominated by 190 fine silt and clay (Table 3).





**Table 2. Soil carbon, nitrogen, molar C:N, isotopic composition ($\delta^{15}$N and $\delta^{13}$C) and bulk density of six reference plots paired with six recently disturbed plots by feral pigs. Reference and disturbed plots were similar for all parameters except for $\delta^{13}$C, which was significantly lower in the disturbed plots ($F_{1,24}$ = 8.66, $p$ = 0.007). Significant differences are shown as $p < 0.05$* and $p < 0.001$**.**

| | C (%) | | N (%) | | C:N | |
|---|---|---|---|---|---|---|
| | Reference | Disturbed | Reference | Disturbed | Reference | Disturbed |
| Ubirr | 9.7 ± 0.5 | 11.6 ± 2.5 | 0.9 ± 0.1 | 0.9 ± 0.1 | 9.7 ± 0.3 | 10.7 ± 0.7 |
| Munmarlay | 7.4 ± 0.8 | 8.2 ± 1.8 | 0.6 ± 0.1 | 0.7 ± 0.1 | 9.8 ± 0.2 | 9.9 ± 0.7 |
| Mamukala | 6.3 ± 0.8 | 4.2 ± 0.5 | 0.4 ± 0.0 | 0.3 ± 0.0 | 15.0 ± 1.4 | 13.1 ± 0.6 |
| Coolaboo | 8.3 ± 1.0 | 6.1 ± 0.5 | 0.7 ± 0.1 | 0.5 ± 0.0 | 9.6 ± 0.2 | 10.1 ± 0.0 |
| Dja Dja | 12.3 ± 1.5 | 13.2 ± 0.9 | 0.8 ± 0.1 | 0.9 ± 0.1 | 12.9 ± 0.3 | 12.5 ± 0.2 |
| Djabulukgu | 4.3 ± 0.3 | 7.8 ± 0.3 | 0.4 ± 0.0 | 0.8 ± 0.0 | 10.3 ± 0.4 | 8.6 ± 0.1 |
| | $\delta^{13}$C (‰) | | $\delta^{15}$N (‰) | | Bulk density (g cm$^{-3}$) | |
| | Reference | Disturbed | Reference | Disturbed | Reference | Disturbed |
| Ubirr | -17.9 ± 1.5 | -17.4 ± 0.4 | 1.2 ± 0.5 | 2.2 ± 0.7 | 0.30 ± 0.02 | 0.27 ± 0.03 |
| Munmarlay | -23.4 ± 0.6 | -21.1 ± 0.9* | 3.6 ± 0.5 | 2.6 ± 0.6 | 0.50 ± 0.04 | 0.45 ± 0.05 |
| Mamukala | -28.8 ± 0.5 | -26.7 ± 0.7 | 3.0 ± 0.5 | 3.2 ± 0.5 | 0.56 ± 0.06 | 0.79 ± 0.08 |
| Coolaboo | -21.1 ± 0.6 | -18.1 ± 0.8** | 1.8 ± 0.1 | 3.2 ± 0.3 | 0.35 ± 0.02 | 0.55 ± 0.02 |
| Dja Dja | -25.3 ± 0.2 | -26.1 ± 0.4 | 1.7 ± 0.6 | 2.9 ± 0.1 | 0.63 ± 0.02 | 0.39 ± 0.03 |
| Djabulukgu | -19.2 ± 0.5 | -19.0 ± 0.2 | 5.1 ± 0.5 | 2.3 ± 0.2 | 0.44 ± 0.02 | 0.27 ± 0.01 |

**Table 3. Soil grain size composition (%) of six reference plots (*R*) paired with six recently disturbed plots by feral pigs (*D*). Bold values highlight the dominant grain size per plot (reference/disturbed) and site.**

| | Gravel | | Fine, medium, coarse sand | | Very fine sand | | Medium and coarse silt | | Fine silt and clay | |
|---|---|---|---|---|---|---|---|---|---|---|
| | *R* | *D* | *R* | *D* | *R* | *D* | *R* | *D* | *R* | *D* |
| Ubirr | 4.0 | 0.9 | 3.8 | 4.9 | 29.0 | 26.4 | **35.6** | 19.9 | 27.5 | **47.9** |
| Munmarlay | 1.2 | 1.2 | 8.8 | 28.9 | 10.5 | 6.0 | 15.0 | 13.7 | **64.4** | **50.1** |
| Mamukala | 6.9 | 6.5 | **61.3** | **63.7** | 11.0 | 7.8 | 10.8 | 6.1 | 10.0 | 15.9 |
| Coolaboo | 5.3 | 0.0 | 17.2 | 1.8 | 25.4 | 0.0 | 17.2 | 2.4 | **35.0** | **95.8** |
| Dja Dja | 4.2 | 1.2 | **46.2** | **51.6** | 15.5 | 12.7 | 11.0 | 17.0 | 23.1 | 17.5 |
| Djabulukgu | 12.4 | 0.5 | **55.3** | 31.9 | 7.5 | 18.5 | 6.7 | 9.9 | 18.1 | **39.2** |






### 3.2 Greenhouse gas emissions

### 3.2.1 Impact of pig disturbance

During Yekke or the cool season, $CO_2$ emissions were significantly different among sites ($F_{5,48}$ = 3.871 $p$ = 0.005) and slightly, but not significantly, lower at the reference (mean of the six sites of 462 ± 101 (286 to 921) mg m$^{-2}$ hr$^{-1}$) compared to disturbed

plots (523 ± 43 (468 – 743) mg m$^{-2}$ hr$^{-1}$; $F_{1,48}$ = 2.87 $p$ = 0.097; Fig.3A). Fluxes of $CH_4$ were significantly different among treatments ($W$ = 248, $p$ = 0.0037, $n$ = 29) with emissions being higher at disturbed plots (663 ± 740, 54 to 4,820 µg m$^{-2}$ hr$^{-1}$) compared to reference plots (375 ± 292, -8.6 to 1,785 µg m$^{-2}$ hr$^{-1}$) at all sites except Coolaboo (Fig. 3B), where the reference plot was not a *Melaleuca* forest but a grass-dominated wetland. The most significant differences were observed for $N_2O$, with higher emissions at disturbed plots ($F_{1,48}$ = 41.5 $p$ < 0.01) with a mean of 81 ± 88.7 (26.6 to 548) µg m$^{-2}$ hr$^{-1}$, which were up to

an order of magnitude higher than those for the reference sites with 11.9 ± 3.0 (2.7 to 20.4) µg m$^{-2}$ hr$^{-1}$ (Fig. 3C).

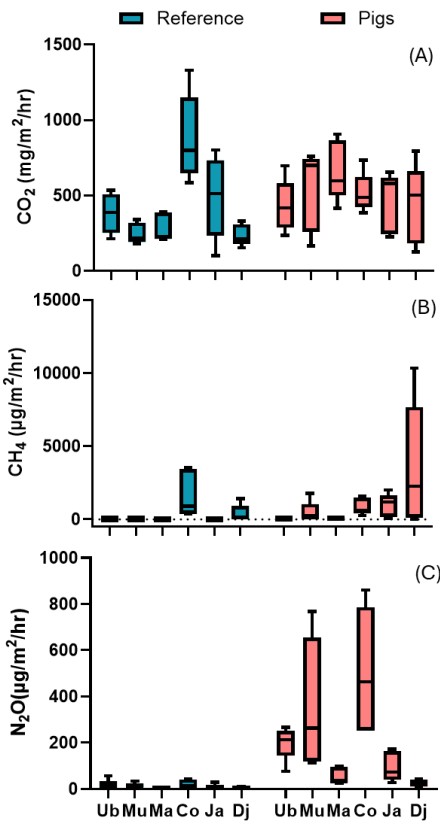

**Figure 3. Greenhouse gas emissions ($CO_2$, $CH_4$ and $N_2O$) from reference plots (blue-left) and plots disturbed by feral pigs (red-right) for six sites across the floodplains in Kakadu National Park: Ub = Ubirr, Mu = Munmarlay, Ma = Mamukala, Co = Coolaboo, Ja = Djaja, Dj = Djabulukgu. Values of the box plots represent the mean and standard error; whiskers represent the minimum and**

**maximum values.**



### 3.2.2 Variability from the time of disturbance

The time since soil disturbance, whether the plot was old or recently used by pigs, was assessed during Kunumleng or the pre-monsoon season. At this time, $CO_2$ emissions were highest at the reference plots, compared to old and new disturbed plots

($F_{2,36} = 6.84$ $p = 0.003$; Fig. 4A). The time of disturbance significantly affected the $CH_4$ emitted, with higher emissions at the newly disturbed plots ($F_{2,36} = 3.60$ $p = 0.038$) compared to old plots or reference sites (Fig. 4B). Finally, $N_2O$ emissions were similar among treatments ($F_{2,36} = 1.07$ $p = 0.354$). However, there was a trend of $N_2O$ uptake (negative values) at the reference plots compared to emissions at the old and disturbed plots (Fig. 4C).

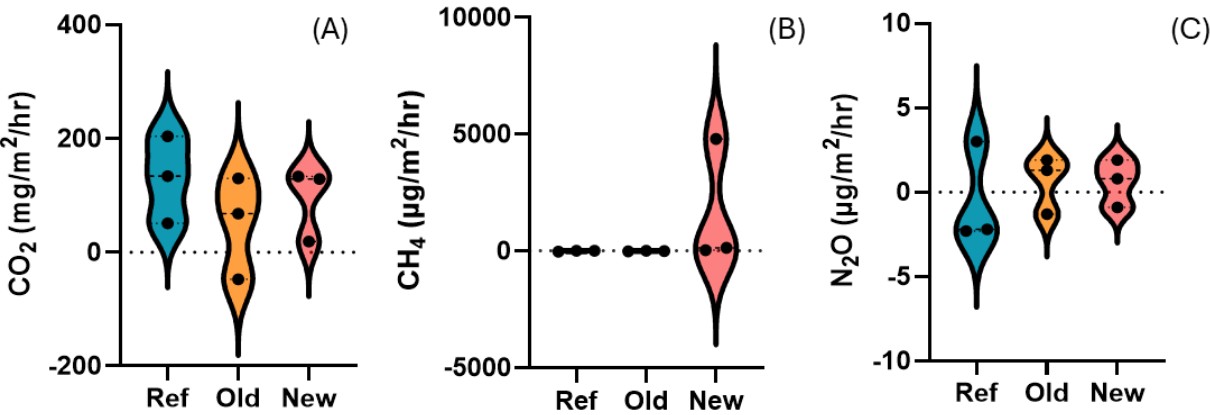


**Figure 4. Greenhouse gas ($CO_2$, $CH_4$, and $N_2O$) fluxes of reference sites (Ref- blue), sites previously used by pigs (Old- orange), and recently used by pigs (New-red). Violins represent the distribution of the mean fluxes (five chambers) within each treatment at three sites (Djabulukgu, Mamukala and Munmarlay) during Kunumleng (pre-monsoon) season.**

### 3.2.3 Variability between seasons

The $CO_2$ emissions were significantly lower during the Kunumeleng (pre-monsoon) season compared to the Yekke (cool) season ($F_{1,8} = 42.29$ $p = 0.0002$; Fig. 5A). The $CH_4$ emissions were similar ($F_{1,13} = 00.148$ $p = 0.71$) but $N_2O$ emissions were significantly lower during Kunumeleng at the disturbed sites ($F_{1,14} = 8.10$ $p = 0.013$; Fig. 5B, C). Redox potential in the disturbed plots was lower during Kunumleng (mean of three sites of $12.2 \pm 44.3$ mV) than during Yekee (mean of five sites of

$67.7 \pm 15.5$ mV).



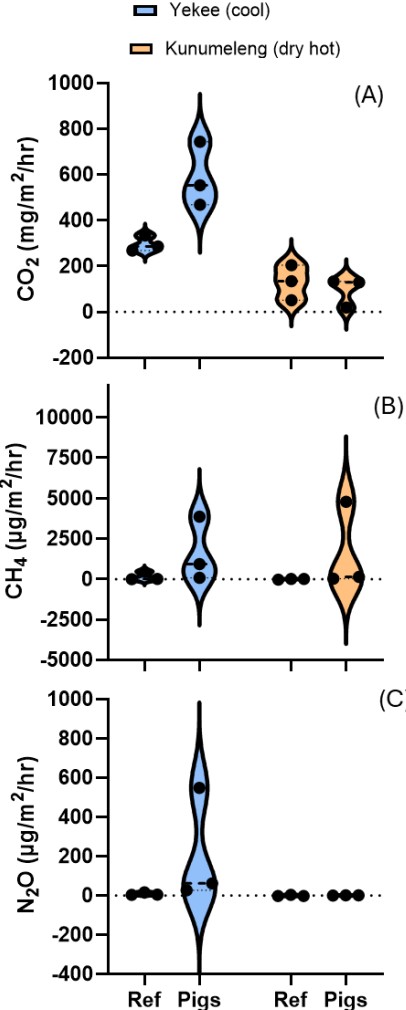

**Figure 5. Greenhouse gas ($CO_2$, $CH_4$ and $N_2O$) fluxes for two seasons: Yekke or cool season (blue-left) and Kunumleng or pre-monsoon season (orange-right), for three plots without (Ref) and with feral pig disturbance (Pigs). Violins represent the distribution of the mean fluxes for five chambers per treatment and three sites (Djabulukgu, Mamukala and Munmurlay).**

### 3.2.4 Physicochemical factors associated with emissions

Only two parameters analysed strongly predicted greenhouse gas fluxes: pig disturbance and soil redox. Soils disturbed by pigs had lower redox than reference or old disturbed plots. Low redox values were correlated with high $CH_4$ emissions, and although the correlation was significant, the regression was poorly fitted to a linear trend (Fig. 4; $R^2 = 0.087$ $p = 0.012$).



Additionally, redox values between 0-100mV were associated with increased $N_2O$ emissions (Fig. 6B), a response that was best fitted to a Gaussian curve; however, the fit was modest and had low statistical confidence ($R^2$ =0.21 $n$ =45, Fig. 6). Nevertheless, a clear pattern was observed between redox and emissions; anaerobic soil with redox values lower or close to zero promoted $CH_4$ emissions and soils with values between 0-100 mV promoted $N_2O$ emissions, but only in plots disturbed by pigs (Fig. 6).

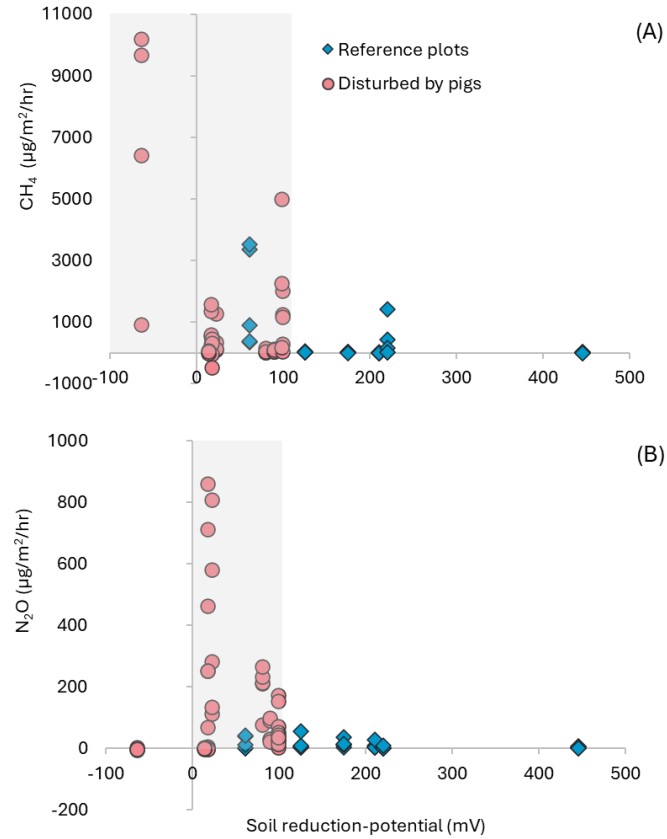

**Figure 6. Relationship between soil redox (reduction-potential, mV) and (A) CH4 and (B) N2O fluxes (µg m$^{-2}$ hr$^{-1}$) from reference (blue diamonds) and disturbed (red circles) plots by feral pigs. Lower redox values were associated with higher CH4 emissions, and values between 0 and 100 mV were associated with higher N2O emissions, but only at plots disturbed by pigs.**

### 3.2.5 Root versus soil respiration

Root biomass was variable, but on average, higher at the reference plots compared to those disturbed by pigs (84 ± 63 vs 17 ± 3 mg cm$^{-3}$; Table 3). Root respiration was similar or even higher than soil respiration, suggesting that most of the soil $CO_2$ emissions in the reference plots originated from root respiration (Table 4). This assumption was further supported when




comparing vegetated and unvegetated plots during the Kunumleg season, with 61% higher emissions in the vegetated plots
(130 ± 44 (51-204) mg m⁻² h⁻¹) compared to the unvegetated plots (50 ± 52 (-47 to 129) mg m⁻² h⁻¹).

**Table 4. Root biomass, root respiration, and soil respiration of reference and disturbed plots by feral pigs.**

| | Root biomass (mg cm⁻³) | | Root respiration (mg m⁻² hr⁻¹) | | Soil respiration (mg m⁻² hr⁻¹) | |
|---|---|---|---|---|---|---|
| | *Reference* | *Disturbed* | *Reference* | *Disturbed* | *Reference* | *Disturbed* |
| Ubirr | 26.4 ± 0.7 | 12.9 ± 8.1 | 2,953 ± 546 | 802 ± 195 | 1,000 ± 49 | 369 ± 58 |
| Munmarlay | 22.9 ± 0.8 | 24.8 ± 13.1 | 501 ± 7.7 | 747 ± 114 | 300 ± 66 | 351 ± 48 |
| Coolaboo | 337 | 18.5 | 2,293 | 1,799 ± 580 | 203 ± 1 | 363 ± 1 |
| Djadja | 16.5 | n.a. | 2,808 ± 546 | n.a. | 1,066 ± 302 | n.a. |
| Djabulukgu | 14.6 ± 4.4 | 13.5 ± 8.5 | 1,193 ± 286 | 1,073 ± 322 | 2,062 ± 96 | 993 ± 355 |

### 3.2.6 Emission reduction potential

The potential reduction of greenhouse gas emissions through feral pig management was estimated per site during sampling in Yekke, where the most spatially comprehensive sampling occurred. We found that avoiding soil disturbance could result in potential $CO_2$ reductions (corrected for root respiration by 61%) at all six sites, $CH_4$ reductions in five out of six sites, and $N_2O$ reductions at all sites (Table 3). When transforming the gases by their warming potential ($CO_{2-eq}$), reductions were highest for $CO_2$ with 33 ± 240 $MgCO_{2eq}$ ha⁻¹ yr⁻¹, followed by $N_2O$ with 2.0 ± 0.2 $MgCO_{2eq}$ ha⁻¹ yr⁻¹ and $CH_4$ with 1.7 ± 1.7 $MgCO_{2eq}$ ha⁻¹ yr⁻¹ (Table 5).



**Table 5. Greenhouse gas emissions (CO₂, CH₄, and N₂O) and emission reduction potential from reference plots compared to disturbed plots by pigs in six sites in Kakadu NP. Values are mean ± standard error for five chambers within each plot and median ± propagated error for the six sites. The soil CO₂ emissions were corrected for root respiration (CO$_{2\text{-resp}}$), which contributed approximately 61% of the emissions.**

| | CO$_{2\text{-resp}}$ | | CH₄ | | N₂O | |
| | mg m⁻² h⁻¹ | | µg m⁻² h⁻¹ | | µg m⁻² h⁻¹ | |
| | Reference | Pigs | Reference | Pigs | Reference | Pigs |
|---|---|---|---|---|---|---|
| Ubirr | 161 ± 25 | 463 ± 93 | 14.1 ± 7.3 | 53.5 ± 25.1 | 17.6 ± 10.9 | 210 ± 32.6 |
| Munmarlay | 111 ± 11 | 592 ± 117 | 5.2 ± 6.9 | 394 ± 225 | 15.7 ± 5.5 | 384.7 ± 135 |
| Mamukala | 129 ± 18 | 743 ± 93 | 0.4 ± 4.9 | 79.1 ± 23.0 | 4.2 ± 1.6 | 61.8 ± 18.3 |
| Coolaboo | 355 ± 49 | 554 ± 56 | 1,785 ± 751 | 932 ± 283 | 20.4 ± 9.6 | 548 ± 132 |
| Djaja | 209 ± 52 | 491 ± 99 | -8.6 ± 4.6 | 1,024 ± 385 | 10.6 ± 5.5 | 100 ± 29.1 |
| Djabulukgu | 104 ± 14 | 468 ± 124 | 453 ± 295 | 4,819 ± 2,160 | 2.7 ± 2.0 | 26.6 ± 7.8 |
| Median ± SE | 178 ± 39 | 523 ± 43 | 10 ± 292 | 663 ± 740 | 11.9 ± 3.0 | 155 ± 84 |
| MgCO$_{2eq}$ ha⁻¹ yr⁻¹ | 13 ± 7 | 46 ± 21 | 0.02 ± 1.9 | 1.6 ± 5.3 | 0.3 ± 0.4 | 3.7 ± 4.7 |
| Emission reductions (by difference) | | | | | | |
| MgCO$_{2eq}$ ha⁻¹ yr⁻¹ | 33 ± 23 | | 1.5 ± 5.6 | | 3.4 ± 4.7 | |
| TOTAL emission reductions | | | | | | |
| MgCO$_{2eq}$ ha⁻¹ yr⁻¹ | **38 ± 24** | | | | | |

## 4 Discussion

The iconic wetlands of Kakadu National Park have been managed for millennia by the Bininj and Mungguy peoples. However, they currently face multiple challenges, including rising sea levels, erratic rainfall patterns, and the impacts of invasive species (Bradshaw et al., 2007b; Pettit et al., 2018). In this study, we found that beyond the known impacts of feral pigs on biodiversity (Risch et al. 2021), they also increase soil CO₂, CH₄, and N₂O emissions, confirming that feral pigs pose a serious threat to soil carbon (O'Bryan et al., 2021). This study provides another compelling example of animal populations significantly affecting the carbon cycle at the landscape scale (Schmitz et al., 2014)

The effect of feral pigs was evident and widespread at all sites, where the soil was turned over and mostly devoid of vegetation. The disturbed plots were also colder and filled with fresh groundwater, as pigs used some of these areas to wallow or coat themselves with mud to avoid overheating (Bracke, 2011). The soil redox potential at all disturbed plots was significantly





reduced, indicating low oxygen conditions. Low oxygen or anoxic soils favour microbial respiration through nitrification-denitrification when nitrate ($NO_3^-$) is available. This process produces $N_2O$ as a byproduct, especially when the soil is high in

nitrogen, such as in soils enriched with pig excreta (Krull et al., 2013). However, soil nitrogen concentrations in our plots were similar in reference and disturbed plots, suggesting that nitrogen enrichment from feral pigs was sporadic and was rapidly removed, potentially through denitrification, which can result in $N_2O$ emissions. Due to low oxygen and potentially high nitrogen loads, the emissions of $N_2O$ during the Yekke season were up to an order of magnitude higher than those in the reference plots.


As the dry season progresses, the weather becomes hotter, and the soil becomes drier. Consequently, pigs move closer to the waterhole fringes to exploit resources at the edge of the open water (Froese et al., 2017). The wallows left behind were highly anaerobic, with redox values close to or below zero (mV). The lack of oxygen in the soil during this time of the year resulted in high $CH_4$ emissions, but low $N_2O$ emissions. This result can be explained by the thermodynamics of respiration; nitrogen is

more energetically favoured than carbon, but once nitrogen is exhausted and oxygen is depleted, methanogenesis becomes the dominant respiration pathway (Reddy & DeLaune, 2008). The disturbed soils showed slightly reduced $^{13}\delta C$ values, indicating methanogenesis. Methane has a negative isotopic value of around -60‰; thus, when produced, it significantly lowers the $^{13}\delta C$ value of the soil (Fry, 2006).

Once the disturbed site was completely dried up and devoid of vegetation, the carbon and nitrogen from the soil were exhausted, and fluxes decreased (Fig. 7). At this point, the $CO_2$ emissions reached their minimum values because $CO_2$ is the net result of two processes: root respiration and soil organic matter decomposition. During the Yekke season, reference sites were fully vegetated and had large amounts of litter accumulated on the soil surface (See Fig. 2A); thus, root respiration and litter decomposition caused high $CO_2$ emissions. The disturbed plots had low vegetation cover and little standing litter, but

likely had high decomposition rates, arising from dead root material and exposed soil organic matter, resulting in similar or higher $CO_2$ emissions compared to the reference plots. During the Kunumleng season, the plots had dried up and had little vegetation or standing litter, resulting in the lowest $CO_2$ emissions measured. Finally, we expect that during Kudjewk, or the monsoon season, the floodplains will overflow again, providing fresh habitat for feral pigs, which will continue to disturb the soil and release the carbon accumulated during the wetter months (Fig. 7).





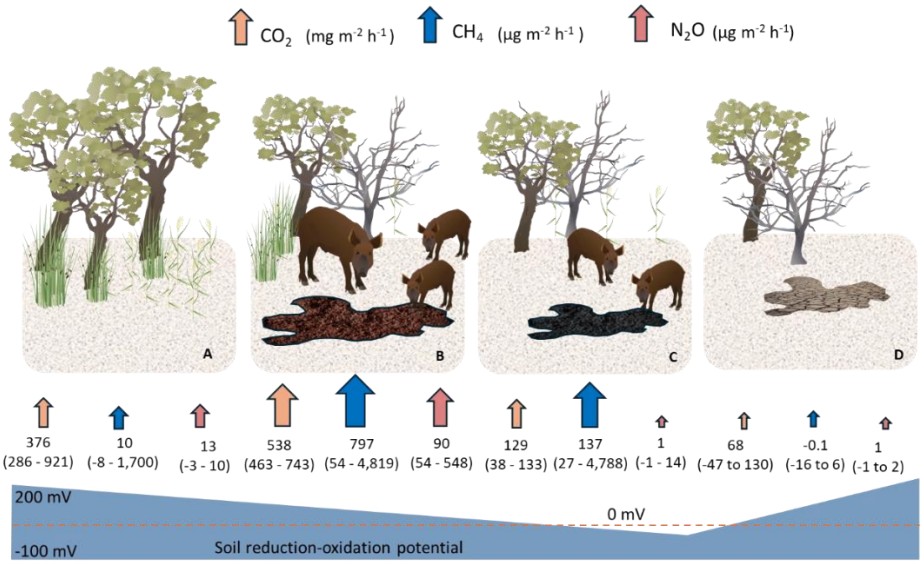


**Figure 7. Schematic diagram on the greenhouse gas emission changes in a floodplain wetland (A) before pig disturbance; (B) during pig disturbance as vegetation is lost, soil redox decreases, and soil decomposition and nitrogen inputs increase, resulting in very high CO₂, CH₄ and N₂O emissions; (C) as the site is less used as animals move closer to the remaining waterholes during drier months, N₂O emissions decrease but CH₄ emissions remain high as oxygen is very low; (D) finally, once the site is abandoned, the soil is depleted of carbon and nitrogen, redox potential increases, and fluxes are minimal. The red hashed line represents oxido-reduction potential values of zero; values close to or below this line significantly increase CH₄ emissions.**


Despite the strong evidence of increased emissions from pig disturbance in this study, we acknowledge some limitations. First, greenhouse gas emissions could not be measured during the wettest months due to the hazardous conditions at the location, 330 particularly the presence of saltwater crocodiles. Emissions during these wet periods could be higher than our measurements, although seasonal differences in greenhouse gas emissions in tropical Australia tend to be relatively low (Iram et al. 2022). Second, the contribution of root versus soil respiration remains unresolved, given the confounding effect of $CO_2$ production from root respiration and soil decomposition (O'Bryan et al. 2022). We used an approximate value of 62% contribution of root respiration to our measurements, which is consistent with other studies (79%, Krauss et al. 2012), although it does not 335 incorporate the potential absorption of $CO_2$ as it travels from the sediment to the atmosphere. However, future studies could refine this estimation, facilitating the inclusion of $CO_2$ in carbon emissions from soil disturbance.

In conclusion, we demonstrate that feral pigs pose a significant threat to the integrity of soil carbon in tropical floodplains in Australia, and likely globally. The physical activity on the soil and the increase in nitrogen through their excreta are 340 significantly increasing emissions, especially of $CH_4$ and $N_2O$. Tackling these two potent greenhouse gases has the potential to significantly contribute to climate change mitigation (Ocko et al., 2021), while also improving biodiversity and human health.



**Author contributions**

MFA, CS, JN, VH, and CEL designed the study; MFA, AP, JMH, JT, CS, and JN carried out field experiments; MFA and JH
analysed the data; MFA wrote the original manuscript, and all co-authors contributed to the editing, interpretation, and
conclusions.

**Competing interests**

The contact author has declared that none of the authors has any competing interest.

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
