# Peer review of "Soil disturbance in wetlands by feral pigs increases greenhouse gas emissions"

_EGUsphere, 2025_

## Referee Comment (RC1)

**General comments**

In regions that take climate change mitigation seriously, research on ecosystem carbon cycling and greenhouse gas GHG emissions is of high relevance, with soils, particularly nutrient-rich soils with high carbon content, being one of the key focuses for nature-based climate stakeholders. By investigating soil GHG emissions, this paper contributes to filling existing knowledge gaps on this matter. The study is particularly relevant by addressing less studied drivers of soil emissions, specifically the impact of feral pigs, thereby providing insight into a less explored aspect of ecosystem functioning. The results offer valuable empirical evidence demonstrating that feral pigs significantly affect soil conditions and GHG emissions, drawing attention to an important issue.

The introduction of the paper successfully and concisely establishes the topicality of the study, provides useful context on the role of invasive ungulates such as feral pigs in tropical Australian ecosystems, and very briefly introduces current understanding of their impacts on GHG emissions. However, the overview of existing knowledge on soil GHG emissions appears somewhat disproportionate and could be slightly expanded.

The amount of sampled data is not explicitly stated and appears relatively limited compared to typical soil GHG emission studies. Nevertheless, this limitation is justified by the specific study conditions, including the remoteness of the area and its inaccessibility or safety constraints during large parts of the year. These constraints do not detract from the overall quality of the study or manuscript. The paired study design, with parallel measurements at disturbed and relatively undisturbed sites, is appropriate and allows for a meaningful assessment of disturbance effects.

The results are sufficient to support the interpretations and conclusions, and the applied methods are generally valid. However, transparency of the Methods section should be improved, particularly by providing more detailed explanations of underlying assumptions, site conditions, and technical aspects of data collection and processing. This would enhance traceability and reproducibility.

Overall, the study is well presented, clearly structured, and written in fluent language. The title accurately reflects the content of the paper.

**Specific comments**

25: The abstract states that animals significantly affect the "carbon cycle"; however, the information presented in the abstract and results indicate that impacts extend beyond the carbon cycle to greenhouse gas emissions more broadly. This could be specified more precisely.

30: "climate change and human impacts", climate change is attributable to human activities; the wording can be adjusted accordingly.

30: "one such solution is the restoration", while restoration is strongly promoted as a climate mitigation measure, empirical evidence remains equivocal, particularly when accounting for spatial and temporal dimensions of uncertainties. The sentence can be adjusted accordingly.

35: "These potent greenhouse gases are emitted in large quantities in wetlands from tropical climates", this is observed in all climates and can be adjusted accordingly.

75: "increase in soil decomposition", the implied linkage between decreasing oxygen availability and increased soil decomposition appears counterintuitive. The sentence may be split into separate sentences addressing different gases for clarity, if necessary.

120: The soil horizon or sampling depth (cm) used for soil sample collection should be specified.

135: The equation 1 appears to be derived from the ideal gas law; therefore, the temperature used should correspond to the chamber headspace (gas) temperature rather than soil temperature. If headspace temperature was not measured, please state this explicitly and discuss the potential uncertainty associated with using soil temperature as a proxy. Soil depth for temperature measurements is not specified.

145-150: Section 2.4. Root sampling depth is not specified. In disturbed plots, measured root-derived emissions likely reflect decomposition processes rather than active respiration. A more accurate approach would therefore be to define these emissions as decomposition rather than respiration. In addition, it is unclear how roots (both dead and living) collected at the reference plots were interpreted in the analysis, whether all measured emissions were assumed to represent decomposition and comparable with estimates from disturbed plots. Furthermore, as soil flux measurements were also conducted at reference sites with undisturbed, vegetated soil, it remains unclear how aboveground autotrophic respiration was treated in the study.

155: "Fluxes of $CO_2$ were also included by correcting for respiration using the differences in emissions (%) from vegetated and unvegetated plots". The purpose of this data manipulation remains unclear at this point in the manuscript. Clarification is required regarding which fluxes were corrected and whether the correction was applied to the

reference plots to account for the presence of living biomass. It is not straightforward to correct part of the dataset by calculating differences between two datasets that are not directly comparable. Specifically, disturbed plots include fluxes from soil organic matter decomposition and decomposition of remaining dead root biomass, whereas reference plots include soil organic matter decomposition, decomposition of roots from natural mortality, respiration of living roots, and respiration of aboveground vegetation. Hence, the purpose of the correction is not clear. Further clarification is therefore needed as to why a simple difference between reference and disturbed plots was not used, and, if root respiration and/or root decomposition emissions were measured, why these data were not applied in the correction. At line 265, it becomes clear that the correction was applied to exclude root emissions; however, the correction procedure should be explained more clearly earlier in the manuscript (section 2.5).

155: The manuscript notes that instantaneous fluxes are extrapolated to annual fluxes per hectare; however, the study design does not allow for accurate annual extrapolation. This should therefore be avoided, or the formulation should be reframed to clearly indicate that the reported values represent instantaneous fluxes expressed in units of t $ha^{-1}$ $yr^{-1}$, with explicit clarification of whether these were obtained by averaging the measurements. An alternative approach could be to present relative reductions in emissions without implying that these can be extrapolated to annual values.

230-235: It is unclear what exactly is being compared. Please clarify whether all data are aggregated or whether the comparison still retains subgrouping by disturbance.

205; 220;230: $CO_2$ emissions were lower at reference plots during Yekke (8–14 June 2023) but higher during Kunumleng (25–30 October 2023). In addition, $CO_2$ emissions during the cooler Yekke period were higher than during the hotter, pre-monsoon Kunumleng period. Can this be explained by seasonal vegetation dynamics? It may be so if by the Kunumleng period, vegetation at reference plots may have developed greater biomass, leading to increased autotrophic respiration and consequently higher $CO_2$ emissions. If so, this would imply that the observed differences reflect not only soil processes but also seasonal variation in vegetation activity. If this is plausible, adding some information on vegetation dynamics may be valuable. At the same time, can the lower $CO_2$ emissions observed during Kunumleng, despite higher temperatures, be related to drier soil conditions, which can limit microbial activity and soil respiration (this was confirmed at discussion)? In this case, adding information on meteorological conditions during the study periods would therefore improve the interpretation of the observed $CO_2$ dynamics.

260: It is unclear which study period is represented by soil respiration in Table 4. Unclear why values in the table do not seem to match those represented in Figure 5. In addition, it is not explained in the methods how measured root emissions (probably "respiration+decomposition" or "decomposition" instead of "respiration" only) are expressed in mg $m^{-2}$ $hr^{-1}$.

335. This part of the sentence is confusing: " although it does not  incorporate the potential absorption of CO2 as it travels from the sediment to the atmosphere."

The hypothesis was not explicitly addressed in the results or discussion.

**Technical corrections**

20: N2O typo.

135-140: multiple typos, format as superscript where necessary.

Use of terms "plots" and "sites" can be harmonised through the manuscript.